# OCT Angiography Features in Diabetes Mellitus Type 1 and 2

**DOI:** 10.3390/diagnostics12122942

**Published:** 2022-11-25

**Authors:** Giovanni William Oliverio, Alessandro Meduri, Gabriella De Salvo, Luigi Trombetta, Pasquale Aragona

**Affiliations:** 1Department of Biomedical Science, Ophthalmology Clinic, University of Messina, 98124 Messina, Italy; 2Ophthalmology Department, University Hospital Southampton, Southampton SO16 5YA, UK

**Keywords:** OCT-angiography, vessel density, FAZ, diabetes mellitus type 1, diabetes mellitus type 2

## Abstract

*Purpose*: To study the foveal avascular zone (FAZ) and the vessel densities (VD) in diabetic patients using optical coherence tomography angiography (OCT-A) and inner retinal layer changes to compare patients affected by type 1 diabetes (DM1) and type 2 diabetes (DM2). *Methods*: Cross-sectional observational study involving 150 eyes of 150 patients with DM1, and 155 eyes of 155 patients with DM2 with diabetic retinopathy (DR). Retinal nerve fiber layer (RNFL) and Ganglion cell layer (GCL) were evaluated. OCT-A studied both FAZ and VD at the level of the superficial capillary plexus (SCP) and the deep capillary plexus (DCP). *Results*: A statistically significant difference in FAZ area and foveal VD measured at the SCP (*p* < 0.001) was noted between DM1 and DM2 groups when comparing patients with mild and moderate non-proliferative diabetic retinopathy (NPDR), while no differences were found in the severe NPDR and proliferative diabetic retinopathy (PDR) subgroups. Duration of diabetes and stage of DR were directly correlated with enlargement of FAZ area and inversely correlated with foveal VD measured at SCP. RNFL and GCL were not different between DM1 and DM2 groups. *Conclusion*: Changes in OCT-A parameters detected in FAZ area and VD of diabetic patients with different stages of DR may help to predict the risk for progression of the disease.

## 1. Introduction

Diabetic retinopathy (DR) represents one of the most important complications of DM and has been recognized as the leading cause of blindness in working-age populations worldwide [1]. DR is a progressive microangiopathy that if left untreated leads to severe complications such as retinal ischemia, neovascularization, and macular edema [2,3]. Fluorescein angiography (FA) is considered the gold standard in detecting DR. FA involves an intravenous injection of dye, which may lead to various side effects and complications. Moreover, FA can only visualize the superficial capillary plexus (SCP), being located above and therefore masking the deep capillary plexus (DCP) [4]. Optical coherence tomography angiography (OCT-A) is able to detect vascular flow based on the movement produced by the blood cells within the vessels. OCTA measures all the retinal vasculature plexi: superficial, deep, and middle [5,6,7]. Furthermore, OCT-A has been shown to detect early microvascular modification of the foveal avascular zone (FAZ) in DR patients even prior to disease onset [5,6,7]. Previous studies have already established that modification in specific OCT-A parameters such as vessel density and FAZ area are related to the severity of DR [8,9,10,11]. Moreover, inner neural retinal layer changes, including reduced retinal nerve fiber layer (RNFL) thickness and glanglion cell layer (GCL) damage have been shown in DR. However, few studies have evaluated the neural layers and microvascular differences in patients with DM1 and DM2. Therefore, we aim to investigate RNFL, GCL and microvascular differences between DM1 and DM2 with DR and to assess the modifications of structural OCT and OCT-A parameters through the progression of DR.

## 2. Materials and Methods

### 2.1. Study Design

This cross-sectional observational study was performed on 305 eyes of 305 patients affected by DM who underwent OCT-A at the retina service of the University of Messina, between September 2018 and December 2019. The study was conducted according to the Declaration of Helsinki, and the Ethics Committee of the University of Messina approved the study (Protocol No. 78/18). All participants provided written informed consent after the explanation of the study’s nature. Inclusion criteria were age >18 years and a history of diabetes mellitus (type 1 or 2). The exclusion criteria included the presence of diabetic macular edema, any other retinal disease, previous retinal surgery, cataract, primary open angle glaucoma or secondary glaucoma, and any other eye disease impacting vision.

Additionally, to avoid the age effect on microvascular changes and to justify a comparison between DM1 and DM2 patients, two control groups of healthy subjects, matched for age respectively with DM1 and DM2 groups, were established.

Patients were evaluated for cardiovascular risks, including age, sex, smoking status, systolic and diastolic blood pressure, height, weight, body mass index, and diabetes based on the glycated hemoglobin A1c (HbA1c) most recent value. Ophthalmic examination included best-corrected visual acuity (BCVA), measured with the early treatment diabetic retinopathy study (ETDRS) chart, slit lamp examination, fundus examination, intraocular pressure, color fundus photographs, and OCT-A. Clinical assessment of DR severity was determined by two trained ophthalmologists (AM, GWO) using the ETDRS severity scales in no DR, mild non-proliferative DR (NPDR), moderate NPDR, severe NPDR, and proliferative DR (PDR) [12].

### 2.2. OCT-A Imaging

DRI OCT Triton plus (Topcon Medical Systems, Inc, Oakland, NJ, USA) was used for the purpose of this study [13]. An experienced technician acquired OCT-A scans (3 × 3 mm area) with high-quality signal; the images were then analyzed [13]. The tool *caliper area* available in IMAGEnet 6 (version 1.17.9720; Topcon Medical Systems, Inc, Oakland, NJ, USA) software was used to measure the FAZ at the level of SCP and DCP. The measurement was done manually and in square millimeters (mm^2^). Vessel density (VD) for both SCP and DCP (3 × 3 mm OCT-A images) was measured with the automatic software algorithm [14,15]. The automatic segmentation used by the machine was described in detail by Stanga et al. [13] The eye showing the better BCVA was chosen for analysis; in cases of equal BCVA, the eye with higher VD was selected. The two different masked operators (AM, OGW) assessed the qualitative changes at the level of the SCP, but also the FAZ area and the capillary features (loss of capillaries, capillary tortuosity, and crossing vessels). The study achieved an inter-observer agreement superior to 95%.

### 2.3. Structural OCT Imaging

A 3D wide scan (12 mm × 9 mm) was acquired and offered information on both macular and RNFL analysis. Ganglion cell layer plus (GCL+), from nerve fiber layer (NFL)/GCL to inner plexiform layer (IPL)/inner nuclear layer (INL) and RNFL segmentation, was performed in six standard areas: superior (S), nasal superior (NS), temporal superior (TS), inferior (I), temporal inferior (TI), and nasal inferior (NI). Total volume was also determined. An active eye tracker was used to reduce motion artifacts.

### 2.4. Statistical Analysis

SPSS software 22 (SPSS, Inc., Chicago, IL, USA) for Windows was used for statistical analysis. We reported measurable data as mean and standard deviation, while absolute frequency and percentage were used to define categorical variables. The Kolmogorov–Smirnov test was used to evaluate the fitting of the data to a normal distribution. For each parameter, a statistical evaluation between groups was assessed using the Student’s *t*-test for parametric data, the Mann–Whitney U test for non-parametric data, and the Chi-Square test for categorical variables. In subgroup analysis, one-way ANOVA with corresponding post hoc analysis for parametric data and the Kruskall–Wallis test for non-parametric data were performed.

Pearson’s correlation analyses were performed to evaluate the correlation between continuous variables. Differences in the RNFL, GCL+, and OCT-A parameters between the groups were calculated with the Fisher exact test. Multiple regression analysis was performed on the OCT-A parameter predictors of diabetic retinopathy stage.

A statistically significant *p*-value was considered to be <0.05. The sample size was established considering an effect size of 0.30 for the difference between means of two examined groups, with reference to FAZ (our clinically relevant variable); as derived from a previous study, a two-sided significance level of 0.05 and a power of 0.80 were used. It was determined that approximately 140 patients per group would be needed (G-power software, 3.1.9.4 version) to reach an adequate power sample.

## 3. Results

### 3.1. Study Population

A total of 150 eyes of patients (76 males, 74 females) with DM1 and 155 with (84 males, 71 females) with DM2 were enrolled in this study. The mean age of patients with DM1 was 55.3 ± 13.1 years (range, 30–65) and 66.7 ± 7.3 years (range, 52–80) for DM2 patients (*p* < 0.001). The mean duration of diabetes was significantly longer in the DM1 group (*p* < 0.001), and the mean recent HbA1c was higher in these patients (*p* < 0.001). Table 1 shows no differences between the two groups in blood pressure mean. In the DM1 group, 39 patients (26.0%) had mild NPDR, 36 (24.0%) had moderate NPDR, 40 (26.7%) had severe NPDR, and 35 (23.3%) had PDR. In the DM2 group, 26 (16.8%) patients had mild NPDR, 47 (30.3%) had moderate NPDR, 42 (27.1%) had severe NPDR, and 40 (25.8%) had PDR (Table 1). To provide control data, two groups of healthy subjects were established: the first control group comprised 40 subjects (22 males, 18 females), with a mean age 53.2 ± 11.4 years (range, 32–67 years), and was matched with the DM1 group, while the second control group comprised 40 subjects (26 males, 14 females), with a mean age 65.2 ± 9.1 years (range, 53–80 years), and was matched with the DM2 group.

### 3.2. FAZ Area

The mean SCP FAZ was significantly larger in the DM2 cohort compared to the DM1 patients (*p* = 0.006). Both DM1 and DM2 groups exhibited a wider area of SCP FAZ compared to the healthy control groups (*p* = 0.001).

In the sub-group analysis, a statistically significant difference was observed in the SCP FAZ area between DM1 and DM2 groups when comparing patients with mild and moderate NPDR, while no differences were detected between severe NPDR and PDR. No differences were recognized at any stage of DR in the mean DCP FAZ area (Table 2). A significant positive correlation was observed between the mean enlargement of the SCP and DCP FAZ areas and duration of diabetes in both groups (Table 3). In multiple regression analysis, the SCP and DCP FAZ areas correlated with diabetic retinopathy stage (Table 4).

### 3.3. Vessel Density

The mean SCP foveal vessel density was 15.1 ± 4.2% in DM1 and 14.6 ± 5.2% in DM2; these were significantly lower with respect to the healthy control groups. The sub-group analysis showed a statistically significant difference in foveal VD between DM1 and DM2 groups when comparing patients with mild and moderate NPDR. There was no statistical difference in VD between the two groups in the four peri-foveal quadrants (Table 1).

Table 3 highlights an inverse correlation between foveal VD and diabetes duration of DR in both groups.

In multiple regression analysis, foveal VD correlated with diabetic retinopathy stage (Table 4).

### 3.4. Qualitative Analysis

We found a significant difference in capillary loss at the level of SCP between the DM1 and DM2 groups amongst eyes with mild NPDR (*p* = 0.007). However, we observed no differences at any stage of DR in the presence of irregular FAZ, crossing vessels, or capillary tortuosity (Table 5 and Figure 1).

### 3.5. GCL+ and RNFL Layers 

GCL+ and RNFL layers were thinner in diabetic patients than controls (*p* < 0.001). However, comparing DM1 and DM2 groups no statistically significant differences of thickness were observed in the following ETDRS locations (Table 6). Furthermore, in subgroup comparison no differences emerged between DM1 and DM2 groups (Table 7).

Both RNFL and GCL+ total thickness correlated significantly with duration of diabetes in DM1 and DM2 groups, but not with HbA1c (Table 3).

## 4. Discussion

Recent studies have demonstrated the crucial role of the neurovascular unit in the pathogenesis of diabetic retinopathy [15,16]. This complex unit comprises the endothelial cells, pericytes, astrocytes, and microglia in a closer functional connection, allowing inner blood–retina barrier regulation in response to metabolic demands. The progressive metabolic changes in diabetes lead to damage of the neurovascular unit, affecting the microvascular structures and the inner retinal layers [15,16]. Indeed, previous studies have demonstrated the reduction of the RNFL, GCL thickness, and microvascular changes as early biomarkers of diabetic retinopathy [15,16,17].

Many studies have shown how we can analyze the retinal structures both quantitively and qualitatively with OCT and OCT-A.

In fact, a significant progressive loss of the RNFL and the GCL could be observed in patients with DM, which may precede microvascular changes characteristic of DR.

Additionally, OCT-A allows estimation of capillary drop-out and area of VD at the level of the plexi [18,19,20,21,22]. Indeed, previous OCT-A studies have proven the presence of retinal capillary non-perfusion and preclinical remodeling in the FAZ of diabetic patients [23,24,25].

Several elements could influence the microvascular modification seen in the macula in diabetic patients, such as HbA1c level, hyperlipidemia, arterial hypertension, renal impairment, and smoking [26].

Ting et al. showed that a variety of systemic metabolic and vascular risk factors were associated with microvascular changes in the macular area observed with OCT-A [26]. Several studies examining OCT-A parameters in diabetic patients with DR have demonstrated a progressive worsening of the FAZ area and VD with DR progression [26]. Moreover, significant alterations in the DCP layers reported in DR patients might be related to an impairment of the photoreceptor layer and hence poor visual acuity prognosis [27]. Furthermore, recent studies demonstrated that the FAZ area and VD of the DCP predicts DR progression, whereas VD of the SCP predicts diabetic macular edema development [3,28]. However, in the literature, there are few studies directly comparing microvascular changes in the macula of patients affected by DM1 and DM2. In accordance with previous OCT-A studies in DR, our results showed microvascular changes in the macula, which severely worsened with DR progression. These changes were found in both groups (DM1 and DM2) and at both the DCP and SCP. However, the greatest alterations were found in the SCP plexus. Indeed, when comparing DM1 and DM2 patients, the DM2 group showed a significantly larger FAZ area in the SCP (*p* = 0.006). Um et al. demonstrated that amongst DM1 patients, VD changes were observed in only the most severe DR stages, while in DM2 the alterations were apparent from the initial stages, with a slow and gradual worsening with the progression of DR. Furthermore, they also reported significant changes in the DCP. Our findings show a significant positive correlation between the FAZ area, measured at both the SCP and DCP in DM1 and DM2 groups, the stage of DR, and the duration of the disease. Moreover, a significant negative correlation was observed in both groups between VD and the stage of DR. In the DM1 group, there were some modifications in VD in the early stages of DR compared to the DM2 group, although in the later stages, no differences were observed between DM1 and DM2 groups. These differences could be attributed to the differences in clinical manifestation, use of insulin, and metabolic control, or the differing pathogenic mechanisms in the two types of diabetes. DM1 patients are usually younger and have a higher risk of developing DR than DM2 patients as well as a higher risk for a faster rate of DR progression [29]. In fact, the FAZ area and VD are known to be related to aging, and studies on healthy subjects have demonstrated FAZ area enlargement and VD reduction in older people [30].

Furthermore, in accordance with previous studies, our findings revealed a reduction of the RNFL and GCL+ in diabetic patients compared to controls [15]. However, no differences emerged when comparing DM1 and DM2 groups.

Inner retinal changes in diabetics were correlated with disease duration, but not with type of diabetes [31,32].

Indeed, Pierro et al. analyzed both RNFL and GCL thickness in diabetics, and no significant differences were reported when comparing DM1 and DM2 patients [31].

In this study, we highlighted the significance of OCT-A findings between DM1 and DM2 patients with the same stage of disease and similar appearances on retinal examination. To the best of our knowledge, this is the first study that has demonstrated a significantly decreased vessel density in the SCP when comparing DM1 and DM2 patients. Our findings suggest that the OCT-A alterations in the macula of DM1 patients could denote the necessity for closer monitoring, to ensure the timely detection of possible complications related to retinal and macular ischemia.

### Limitations

A limitation of our study is that the OCT-A scans evaluated only a small part of the posterior pole; therefore, a study of the peripheral retina microvasculature is lacking. In addition, the gap in age between the two groups may be a confounding factor in the differences reported between the two types of diabetes. Finally, systemic comorbidities such as hypertension, hyperlipidemia, and renal function should be deeply analyzed to understand their influences on the microvascular retinal structure. Further studies with long-term follow-up of patients are warranted to better elucidate the microvascular changes in the macular area and their significance in the progression of DR.

## 5. Conclusions

In conclusion, OCT-A can identify structural microvascular changes in DM1 and DM2 patients associated with the severity of diabetic retinopathy and may therefore be a promising tool for the screening of diabetic eyes for DR progression.

## Figures and Tables

**Figure 1 diagnostics-12-02942-f001:**
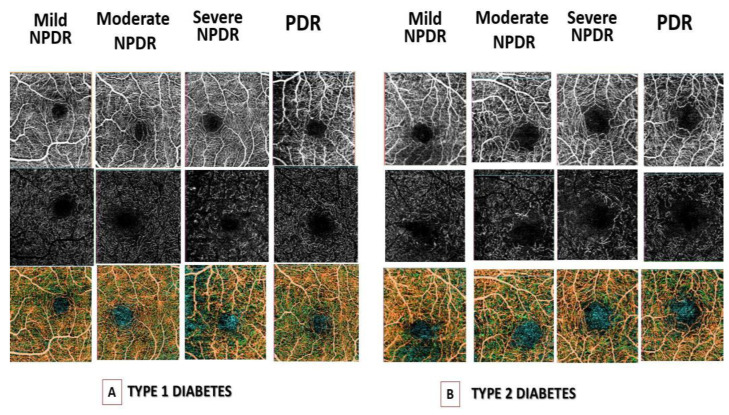
Optical coherence tomography angiography (OCT-A) images of (**A**) Foveal avascular zone (FAZ) area in superficial capillary plexus (SCP) (top row), in deep capillary plexus (DCP) (middle row) and vessel density (VD) (bottom row) in type I diabetes mellitus (DM) patients with different stages of diabetic retinopathy. (**B**) FAZ area in SCP (top row), DCP (middle row) and VD (bottom row) in type II DM patients with different stages of diabetic retinopathy. Legend: SCP-Superficial plexus; DCP-Deep vascular plexus; DR-Diabetic Retinopathy; VD-Vessel Density; NPDR- non-proliferative diabetic retinopathy, PDR diabetic retinopathy.

**Table 1 diagnostics-12-02942-t001:** Clinical characteristics of the study population.

Variables	DM1	DM2	Control Group 1	Control Group 2	*p*-Value
Age (years)	55.3 ± 13.1	66.7 ± 7.3	55.6 ± 10.6	66.5 ± 13.4	**<0.001**
Gender (male/female)	74/76	84/71	22/18	26/14	0.41
Duration of diabetes (years)	20.4 ± 8.1	17.5 ± 8.8	-	-	**<0.001**
HbA1c (%)	7.9 ± 0.8	7.2 ± 0.7	-	-	**<0.001**
Glycaemia	139.1 ± 23.8	140.2 ± 20.4	-	-	0.27
Mild NPDR	39 (26.0%)	26 (16.8%)	-	-	0.11
Moderate NPDR	36 (24.0%)	47 (30.3%)	-	-	0.19
Severe NPDR	40 (26.7%)	42 (27.1%)	-	-	0.69
PDR	35 (23.3%)	40 (25.8%)	-	-	0.38
FAZ area SCP (mm^2^)	0.550 ± 0.12	0.592 ± 0.18	0.271 ± 0.09	0.276 ± 0.11	0.62
FAZ area DCP (mm^2^)	0.690 ± 0.15	0.710 ± 0.15	0.320 ± 0.11	0.307 ± 0.12	0.50
VD (%)	15.1 ± 4.2	14.6 ± 5.2	22.3 ± 5.1	22.9 ± 5.1	0.51
Superior VD (%)	45.6 ± 3.0	45.2 ± 3.4	49.8 ± 5.2	48.1 ± 5.5	0.21
Inferior VD (%)	47.1 ± 3.2	46.3 ± 3.8	48.2 ± 3.9	46.5 ± 4.2	0.13
Nasal VD (%)	45.7 ± 3.7	44.9 ± 2.9	50.1 ± 5.6	50.4 ± 5.2	0.06
Temporal VD (%)	45.1 ± 2.7	45.5 ± 3.7	49.5 ± 4.6	48.7 ± 5.1	0.07

DM1: type 1 diabetes mellitus; DM2: type 2 diabetes mellitus; HbA1c: glycated hemoglobin; FAZ: foveal avascular zone; VD: vessel density; SCPL: superficial plexus; DCP: deep vascular plexus. Bold characters indicate *p*-value < 0.05.

**Table 2 diagnostics-12-02942-t002:** Comparison Between Patients with Type 1 and Type 2 Diabetes Mellitus at Each Diabetic Retinopathy Stage.

Variables	DM1	DM2	*p*-Value *
*FAZ SCP* (mm^2^)			
Mild NPDR	0.408 ± 0.46	0.306 ± 0.52	**<0.001**
Moderate NPDR	0.532 ± 0.34	0.479 ± 0.53.1	**<0.001**
Severe NPDR	0.574 ± 0.60	0.551 ± 0.30	0.17
PDR	0.699 ± 0.72	0.707 ± 0.79	0.23
***p*-value ****	**<0.001**	**<0.001**	
*FAZ DCP* (mm^2^)			
Mild NPDR	0.516 ± 60.3	0.522 ± 0.46	0.77
Moderate NPDR	0.647 ± 30.7	0.622 ± 0.46	0.12
Severe NPDR	0.729 ± 35.4	0.725 ± 0.58	0.79
PDR	0.885 ± 118.2	0.919 ± 0.88	0.14
***p*-value ****	**<0.001**	**<0.001**	
*VD (%)*			
Mild NPDR	20.3 ± 3.9	23.3 ± 3.8	**<0.001**
Moderate NPDR	14.9 ± 0.9	16.6 ± 1.1	**<0.001**
Severe NPDR	13.7 ± 1.9	12.9 ± 1.3	0.91
PDR	10.9 ± 1.8	10.6 ± 1.4	0.76
***p*-value ****	**<0.001**	**<0.001**	
*Superior VD (%)*			
Mild NPDR	44.8 ± 2.1	48.3 ± 2.4	**<0.001**
Moderate NPDR	47.6 ± 1.8	43.4 ± 3.5	**0.03**
Severe NPDR	45.6 ± 3.9	43.9 ± 2.4	0.07
PDR	44.6 ± 2.8	46.6 ± 2.8	**0.02**
***p*-value ****	0.37	0.19	
*Inferior VD (%)*			
Mild NPDR	46.2 ± 3.4	47.7 ± 4.1	0.83
Moderate NPDR	45.7 ± 3.2	45.7 ± 3.2	0.11
Severe NPDR	47.3 ± 2.5	47.9 ± 3.3	0.11
PDR	47.4 ± 3.5	44.5 ± 4.1	**0.002**
***p*-value ****	0.51	0.39	
*Nasal VD (%)*			
Mild NPDR	45.4 ± 4.4	45.1 ± 3.0	0.62
Moderate NPDR	45.6 ± 3.7	44.2 ± 2.8	**0.01**
Severe NPDR	46.2 ± 3.6	44.8 ± 2.8	**0.007**
PDR	45.9 ± 3.4	45.8 ± 2.9	0.56
***p*-value ****	0.68	0.84	
*Temporal VD (%)*			
Mild NPDR	44.6 ± 2.9	47.6 ± 2.8	0.11
Moderate NPDR	45.5 ± 1.5	44.9 ± 3.7	0.42
Severe NPDR	44.6 ± 2.4	45.1 ± 3.9	0.71
PDR	46.1 ± 3.4	45.5 ± 4.0	**0.07**
***p*-value ^**^**	0.21	0.14	

DM1: type 1 diabetes mellitus; DM2: type 2 diabetes mellitus; FAZ: foveal avascular zone; VD: vessel density; SCP: superficial plexus; DCP: deep vascular plexus; NPDR: non-proliferative diabetic retinopathy; PDR: proliferative diabetic retinopathy. * Comparison between DM1 and DM2 groups. ** Multiple comparison between subgroups. Bold characters indicate *p*-value < 0.05.

**Table 3 diagnostics-12-02942-t003:** Correlation between Optical Coherence Tomography Angiography Parameters and Diabetes data.

	DM1					DM2				
Parameter	FAZ SCP	FAZ DCP	VD	RNFL	GCL	FAZ SCP	FAZ DCP	VD	RNFL	GCL
Duration	0.43 ***p* = 0.03**	0.38 ***p* = 0.01**	−0.34 ***p* < 0.001**	−0.28 ***p* < 0.001**	−0.21 ***p* < 0.001**	0.64 ***p* < 0.001**	0.41 ***p* = 0.02**	−0.39 ***p* < 0.001**	−0.26 ***p* < 0.001**	−0.23 ***p* < 0.001**
HbA1C (%)	0.07 *p* = 0.23	0.13 *p* = 0.19	0.11 *p* = 0.1	0.08 *p* = 0.2	0.16 *p* = 0.23	0.18 *p* = 0.32	0.15 *p* = 0.07	0.21 *p* = 0.11	0.19 *p* = 0.16	0.21 *p* = 0.14

**Table 4 diagnostics-12-02942-t004:** Multiple regression analysis on the OCT-A parameter predictors of diabetic retinopathy stage.

Variables	DM1		DM2	
	β	*p*-Value	β	*p*-Value
FAZ SCP	0.003	**0.004**	0.004	**<0.001**
FAZ DCP	0.004	**<0.001**	0.001	**0.03**
VD	−0.026	**0.04**	−0.04	**0.03**

DM1: type 1 diabetes mellitus; DM2: type 2 diabetes mellitus; FAZ: foveal avascular zone; SCP: superficial plexus; DCP: deep vascular plexus; VD: vessel density; HbA1c: glycated hemoglobin. Bold characters indicate *p*-value < 0.05.

**Table 5 diagnostics-12-02942-t005:** Qualitative optical coherence tomography angiography analysis.

	DM1				DM2			
Parameter	Mild NPDR	Moderate NPDR	Severe NPDR	PDR	Mild NPDR	Moderate NPDR	Severe NPDR	PDR
Irregular FAZ	11 (28.2%)	18 (50%)	24 (60%)	28 (80%)	11 (42.3%)	26 (55.3)	25 (62.5%)	33 (78.6%)
Capillary loss	8 (20.5%)	15 (41.7%)	27 (67.5%)	31 (88.6%)	14 ^a^ (53.8%)	24 (51.1%)	30 (75%)	36 (85.7%)
Capillary tortuosity	12 (30.8%)	18 (50%)	31 (77.5%)	33 (94.3%)	11 (42.3%)	28 (59.6%)	31 (77.5%)	38 (90.5%)
Crossing vessel	2 (5.1%)	2 (5.6%)	4 (11.1%)	8 (22.9%)	2 (7.7%)	3 (6.4%)	5 (12.5%)	9 (21.4%)

DM1: type 1 diabetes mellitus; DM2: type 2 diabetes mellitus; NPDR: non-proliferative diabetic retinopathy; PDR: proliferative diabetic retinopathy. Capillary loss, tortuosity, and the presence of crossing vessels were evaluated at the superficial capillary plexus in the macular area. ^a^
*p* = 0.007 (vs. DM1 with mild NPDR).

**Table 6 diagnostics-12-02942-t006:** Structural Optical Coherence Tomography Comparison Between Patients with Type 1 and Type 2 Diabetes Mellitus.

Variables	DM1	DM2	Control Group 1	Control Group 2	*p*-Value
**GCL+**					
**Total**	70.1 ± 1.1	69.7 ± 1.0	75.1 ± 1.2	74.9 ± 1.0	0.31
**Superior**	69.9 ± 1.0	69.5 ± 1.2	73.5 ± 1.1	73.2 ± 1.1	0.44
**Sup. Temporal**	70.8 ± 0.9	70.4 ± 0.8	74.2 ± 0.8	74.1 ± 1.0	0.73
**Sup. Nasal**	72.7 ± 0.7	72.6 ± 0.8	76.8 ± 1.1	76.5 ± 0.9	0.8
**Inferior**	67.4 ± 1.1	67.3 ± 1.0	70.1 ± 0.8	69.9 ± 0.9	0.85
**Inf. Temporal**	71.7 ± 0.9	71.9 ± 0.9	75.1 ± 0.9	74.9 ± 1.0	0.61
**Inf. Nasal**	71.9 ± 0.8	71.9 ± 0.9	75.4 ± 1.1	75.3 ± 0.9	0.89
**RNFL**					
**Total**	106.9 ± 1.5	106.5 ± 1.8	110.8 ± 1.4	110.4 ± 1.3	0.81
**Temporal**	76.3 ± 1.7	76.2 ± 1.5	79.4 ± 1.6	79.6 ± 1.5	0.84
**Sup. Temporal**	139.9 ± 2.8	139.5 ± 2.4	146.8 ± 2.1	146.5 ± 1.8	0.91
**Inf. Temporal**	144.3 ± 2.7	144.1 ± 2.8	150.4 ± 2.5	150.3 ± 2.7	0.92
**Nasal**	90.6 ± 2.2	90.5 ± 2.0	93.5 ± 1.7	93.2 ± 1.9	0.89
**Sup. Nasal**	120.7 ± 3.2	120.6 ± 3.6	121.7 ± 2.9	121.5 ± 3.1	0.79
**Inf. Nasal**	137.1 ± 3.4	136.8 ± 3.1	140.9 ± 3.3	140.7 ± 3.2	0.41

GCL—Ganglion Cell Layer plus; RNFL—Retinal Nerve Fiber Layer; DM1—Type 1 diabetes mellitus; DM2—Type 2 diabetes mellitus. Bold characters for *p*-value < 0.05.

**Table 7 diagnostics-12-02942-t007:** Ganglion Cell Layer and Retinal Nerve Fiber layer Comparison Between Patients with Type 1 and Type 2 Diabetes Mellitus at Each Diabetic Retinopathy Stage.

Variables	DM1	DM2	*p*-Value *
*GCL + Total* (μm)			
Mild NPDR	70.4 ± 1.3	70.1 ± 1.5	0.37
Moderate NPDR	70.6 ± 1.5	70.2 ± 1.3	0.13
Severe NPDR	70.1 ± 1.3	70.4 ± 1.3	0.27
PDR	69.6 ± 1.3	68.9 ± 1.3	0.08
** *p* ** **-value ****	0.13	0.08	
*GCL + Superior* (μm)			
Mild NPDR	70.7 ± 1.5	70.5 ± 1.2	0.22
Moderate NPDR	70.4 ± 1.6	70.4 ± 1.3	0.48
Severe NPDR	69.5 ± 1.7	69.8 ± 1.6	0.35
PDR	68.4 ± 1.5	68.5 ± 1.4	0.38
** *p* ** **-value ****	0.09	0.08	
*GCL + Sup. Temporal* (μm)			
Mild NPDR	71.7 ± 1.2	71.5 ± 1.4	0.75
Moderate NPDR	70.9 ± 1.4	70.7 ± 1.4	0.62
Severe NPDR	70.2 ± 1.4	70.4 ± 1.5	0.59
PDR	69.7 ± 1.3	69.5 ± 1.4	0.49
** *p* ** **-value ****	0.13	0.12	
*GCL + Sup. Nasal* (μm)			
Mild NPDR	74.5 ± 1.5	74.6 ± 1.6	0.72
Moderate NPDR	73.6 ± 1.4	73.3 ± 1.4	0.46
Severe NPDR	71.9 ± 1.5	72.1 ± 1.4	0.32
PDR	69.8 ± 1.5	69.9 ± 1.3	0.88
** *p* ** **-value ****	**0.04**	**0.04**	
*GCL + Inferior* (μm)			
Mild NPDR	70.1 ± 1.2	70.3 ± 1.4	0.77
Moderate NPDR	69.6 ± 1.4	69.7 ± 1.3	0.83
Severe NPDR	68.1 ± 1.4	68.5 ± 1.5	0.79
PDR	65.4 ± 1.4	65.5 ± 1.4	0.88
** *p* ** **-value ****	**0.03**	**0.02**	
*GCL + Inf. Temporal* (μm)			
Mild NPDR	72.6 ± 1.5	72.4 ± 1.6	0.55
Moderate NPDR	72.1 ± 1.5	72.2 ± 1.3	0.51
Severe NPDR	71.4 ± 1.3	71.5 ± 1.3	0.47
PDR	70.2 ± 1.4	70.4 ± 1.3	0.67
** *p* ** **-value ****	0.09	0.08	
*GCL + Inf. Nasal* (μm)			
Mild NPDR	72.4 ± 1.4	72.4 ± 1.5	0.96
Moderate NPDR	71.3 ± 1.5	71.2 ± 1.3	0.83
Severe NPDR	70.4 ± 1.4	70.5 ± 1.3	0.79
PDR	70.1 ± 1.4	70.2 ± 1.4	0.88
** *p* ** **-value ****	0.13	0.11	
*RNFL Total* (μm)			
Mild NPDR	107.6± 1.8	107.5± 1.6	0.89
Moderate NPDR	107.4± 1.5	107.7± 1.5	0.63
Severe NPDR	106.8± 1.9	106.9± 1.7	0.79
PDR	106.2± 1.8	106.6± 1.9	0.55
** *p* ** **-value ****	0.27	0.31	
*RNFL Temporal* (μm)			
Mild NPDR	77.8 ± 1.6	77.6 ± 1.7	0.75
Moderate NPDR	77.2 ± 1.5	77.1 ± 1.6	0.82
Severe NPDR	76.5 ± 1.6	76.4 ± 1.6	0.79
PDR	75.1 ± 1.6	75.2 ± 1.7	0.83
** *p* ** **-value ****	0.14	0.15	
*RNFL Sup. Temporal* (μm)			
Mild NPDR	142.5 ± 2.8	142.6 ± 2.6	0.91
Moderate NPDR	141.8 ± 2.7	141.7 ± 2.6	0.91
Severe NPDR	140.4 ± 2.7	140.5 ± 2.5	0.91
PDR	139.2 ± 2.5	139.3 ± 2.6	0.91
** *p* ** **-value ****	0.13	0.11	
*RNFL Inf. Temporal* (μm)			
Mild NPDR	146.5 ± 2.6	146.4 ± 2.6	0.94
Moderate NPDR	145.4 ± 2.6	145.3 ± 2.6	0.88
Severe NPDR	144.6 ± 2.7	144.6 ± 2.7	0.92
PDR	143.2 ± 2.8	143.3 ± 2.6	0.90
** *p* ** **-value ****	0.07	0.08	
*RNFL Nasal* (μm)			
Mild NPDR	92.3 ± 2.3	92.5 ± 2.2	0.76
Moderate NPDR	91.6 ± 2.3	91.7 ± 2.3	0.81
Severe NPDR	90.4 ± 2.2	90.6 ± 2.3	0.86
PDR	89.6 ± 2.3	89.5 ± 2.3	0.91
** *p* ** **-value ****	0.06	0.07	
*RNFL Sup. Nasal* (μm)			
Mild NPDR	122.6 ± 3.3	122.7 ± 3.2	0.81
Moderate NPDR	121.5 ± 3.2	121.7 ± 3.4	0.72
Severe NPDR	120.8 ± 3.3	120.7 ± 3.3	0.77
PDR	119.4 ± 3.3	119.5 ± 3.4	0.79
** *p* ** **-value ****	0.09	0.08	
*RNFL Inf. Nasal* (μm)			
Mild NPDR	139.6 ± 3.3	139.7 ± 3.3	0.52
Moderate NPDR	138.8 ± 3.4	138.7 ± 3.4	0.53
Severe NPDR	137.6 ± 3.5	137.8 ± 3.5	0.35
PDR	136.2 ± 3.4	136.4 ± 3.4	0.31
** *p* ** **-value ****	0.12	0.11	

GCL—Ganglion Cell Layer plus; RNFL—Retinal Nerve Fiber Layer; DM1—Type 1 diabetes mellitus; DM2—Type 2 diabetes mellitus. * Comparison between DM1 and DM2 groups. ** Multiple comparison between subgroups. Bold characters indicate *p*-value < 0.05.

## Data Availability

Data are available upon request.

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
