# Peer review of "OCT Angiography Features in Diabetes Mellitus Type 1 and 2"

_diagnostics, 2022, doi:10.3390/diagnostics12122942_

Round 1

Reviewer 1 Report

This research article of comparing OCTA features between diabetes mellitus type 1 with 2. This manuscript is over-interpreted and does not have much novelty. The majority of the results have been reported elsewhere. There are also major problems in the experimental design.

1. The main purpose of this article is comparing OCTA features between diabetes mellitus type 1 with 2, but the age mismatch between the two groups(p<0.001). It is well known that retinal vessel geometric gharacteristics changes are affected by age.

2. It is not clear why author set two control group while the main purposes and main results are not related with the control group. Type 1 diabetes and type 2 diabetes can serve as controls for each other. The normal control group seems to be a little superfluous, and it also makes the contents of the table very confusing.

3. The results in Table 6 show that with the development of DR, the change trend of GCL thickness in DM1 group and DM2 group is inconsistent. The author does not explain the reason for this change.

Author Response

Dear Reviewer,

We are grateful to you for your time and constructive comments on our manuscript. We have implemented the manuscript according to your comments. Changes in the last version of the manuscript are reported as highlighted the changes. Below, we also provide a point-by-point response explaining how we have addressed each of your comments.

  1. The main purpose of this article is comparing OCTA features between diabetes mellitus type 1 with 2, but the age mismatch between the two groups(p<0.001). It is well known that retinal vessel geometric gharacteristics changes are affected by age.

We agree with your comments, indeed in limitations we have specified that age has a significant impact on microvessel characteristics. However, this cannot be avoided comparing different stages of diabetic retinopathy in DM type I and type II patients. To the best of our knowledge there are few studies with a large group that compared DMI and DMII patients OCTA features and inner retinal layers.

  1. It is not clear why author set two control group while the main purposes and main results are not related with the control group. Type 1 diabetes and type 2 diabetes can serve as controls for each other. The normal control group seems to be a little superfluous, and it also makes the contents of the table very confusing.

Considering the difference in the mean age of DMI and DMII patients, we decided on two control groups matched for age. This demonstrates that most of the OCTA differences observed are related to the DR.

We agree that the table is confusing with these groups, indeed we have changed it, reporting the main findings in the results section.  

  1. The results in Table 6 show that with the development of DR, the change trend of GCL thickness in DM1 group and DM2 group is inconsistent. The author does not explain the reason for this change.

Thank you for your suggestions, we have improved the discussion section.

Reduction of GCL and RNFL thickness has been reported as early biomarkers of diabetic retinopathy, demonstrating the involvement of the neurovascular unit also in the preclinical forms of DR.

Our findings confirmed a significant reduction of both GCL and RNFL thickness in diabetic patients compared to controls, however, any differences emerged comparing DMI and DMII groups. These findings were in accordance with previous studies that demonstrated early inner retinal changes in diabetics and correlated with the duration of disease, but not with the type of diabetes.

Reviewer 2 Report

Dear authors,

Congratulations for this amazing paper. OCTA is a great tool which offers us new diagnostic possibilities in the very near future, so this kind of studies are crucial.

Here are some of my suggestions and corrections:

Line 36:

OCTA studiesà change by OCTA measures or analyses

Line 48:

All subsections have a name, except 2.1, maybe a title such as “study design” will be useful.

All subsections’ titles are followed by text, so is difficult to identify each one, leave blank after title.

OCTA-imaging:

Why did authors choose the 3*protocol? Is a very small area compared to other protocols available on this device

Line 74:

Please review bibliography, as text index don’t match bibliography, i.e Stanga et al.14 appears an nº 13 on bibliography.

Statistical analysis:

Why did authors perform T-test and Mann-Whitney U test?? Is not the appropriate test for multiple comparisons, Anova test is performed in that case. One way Anova with corresponding posthoc analysis for parametric data and Anova Kruskall Wallis with DCSF pairwaise posthoc for non parametric data.

Table 2:

<0.0001 appears several times, please remove one zero.

Tables 1,2&3:

Please provide 3 decimals even if the last one is zero.

Table 3:

Stage is a qualitative variable even if it’s categorical continuous, 1 (no DR), 2 (NPDR), 3 (moderate NPDR), 4 (severe NPDR), and 5 (PDR). You cannot apply Pearson’s logic to categorical variables

Line 144-145

There was no statistical difference in VD between the two groups in the 4 144 peri-foveal quadrants. Please add that this info is on Table 1.

Qualitative analysis:

We found a significant difference in capillary loss at the level 147 of SCP between the DM1 and DM2 groups amongst eyes with mild NPDR (Table 4).

Please add here p=0.007 instead on table caption, it will be more notable.

Table 6:

GCL + (m) and RNFL (m). Please provide which variable of GCL and RNFL is provided on the table. Is the mean RNFL and GCL? It will be very interesting to analyse all quadrants.

Also, the same correlations provided for OCTA will be very useful for RNFL and GCL (Duration, Stage and HbA1C (%)).

Statistic analysis:

Subanalysis between stage groups in each DM group is very interesting, but it will enrich the paper to compare each stage group to know if there are differences for each parameter in each stage (1 (no DR) vs 2 (NPDR), 1 (no DR) vs 3 (moderate NPDR), 2 (NPDR) vs 3 (moderate NPDR), ….

Did the authors considered performing a logistic regression between FAZ and stage?

Author Response

Dear Reviewer,

We are grateful to you for your time and constructive comments on our manuscript. We have implemented the manuscript according to your comments. Changes in the last version of the manuscript are reported as highlighted the changes. Below, we also provide a point-by-point response explaining how we have addressed each of your comments.

Line 36:

OCTA studiesà change by OCTA measures or analyses

Line 48:

All subsections have a name, except 2.1, maybe a title such as “study design” will be useful.

All subsections’ titles are followed by text, so is difficult to identify each one, leave blank after title.

Many thanks, we included these changes in the text.

OCTA-imaging:

Why did authors choose the 3*protocol? Is a very small area compared to other protocols available on this device

Thank you for this comment. We decided for 3x3mm scan as reported in previous studies. This scan allows us to evaluate the FAZ area for vessel density measurements and morphologic evaluations.

Fenner BJ, et al. Identification of imaging features that determine quality and repeatability of retinal capillary plexus density measurements in OCT angiography. Br J Ophthalmol. 2018 Apr;102(4):509-514. doi: 10.1136/bjophthalmol-2017-310700.

Line 74:

Please review bibliography, as text index don’t match bibliography, i.e Stanga et al.14 appears an nº 13 on bibliography.

Thank you, we have corrected the references numerations as suggested.

Statistical analysis: 

Why did authors perform T-test and Mann-Whitney U test?? Is not the appropriate test for multiple comparisons, Anova test is performed in that case. One way Anova with corresponding posthoc analysis for parametric data and Kruskall Wallis with DCSF pairwaise posthoc for non parametric data.

Many thanks for these comments and suggestions. We performed a paired comparison between DM1 and DM2 groups for each diabetic retinopathy stage. Furthermore, we decided for two control groups matched for age, considering the differences of ages between DM1 and DM2 goups. However, these comparisons were paired and not multiple (es. comparing only DM1 mild NPDR vs DM2 mild NPDR), for these reasons we decided for Student's t-test for parametric data, the Mann-Whitney U test for non-parametric data. We included also a multiple comparison for subgroup analysis using ANOVA test and posthoc analysis.

Table 2:

<0.0001 appears several times, please remove one zero.

Tables 1,2&3:

Please provide 3 decimals even if the last one is zero.

Thank you, we correct these results as suggested

Table 3:

Stage is a qualitative variable even if it’s categorical continuous, 1 (no DR), 2 (NPDR), 3 (moderate NPDR), 4 (severe NPDR), and 5 (PDR). You cannot apply Pearson’s logic to categorical variables

Many thanks for this advice. We removed the correlation analysis with DR stage.

Line 144-145

There was no statistical difference in VD between the two groups in the 4 144 peri-foveal quadrants. Please add that this info is on Table 1.

We made these changes as advised.

Qualitative analysis:

We found a significant difference in capillary loss at the level 147 of SCP between the DM1 and DM2 groups amongst eyes with mild NPDR (Table 4).

Please add here p=0.007 instead on table caption, it will be more notable.

We made these changes as advised.

Table 6:

GCL + (m) and RNFL (m). Please provide which variable of GCL and RNFL is provided on the table. Is the mean RNFL and GCL? It will be very interesting to analyse all quadrants.

Also, the same correlations provided for OCTA will be very useful for RNFL and GCL (Duration, Stage and HbA1C (%)).

We have included all data in table 7, however not significantly differences emerged between two groups. Moreover, both RNFL and GCL thickness correlated with duration of diabetes.

Statistic analysis:

Subanalysis between stage groups in each DM group is very interesting, but it will enrich the paper to compare each stage group to know if there are differences for each parameter in each stage (1 (no DR) vs 2 (NPDR), 1 (no DR) vs 3 (moderate NPDR), 2 (NPDR) vs 3 (moderate NPDR), ….

Did the authors considered performing a logistic regression between FAZ and stage?

 Many thanks for your advice, we included a multiple regression analysis between OCT-A parameters and DR stage in table 4. Moreover, in subgroups we have performed a multiple comparison for each parameter.